# Computational models to improve surveillance for cassava brown streak disease and minimize yield loss

**Alex C. Ferris** [1] *, **Richard O. J. H. Stutt** [2], **David Godding** [2,3], **Christopher A. Gilligan** [2]

**1** Department of Bioengineering, Stanford University, Stanford, California, United States of America,
**2** Department of Plant Sciences, University of Cambridge, Cambridge, United Kingdom, **3** Farming Data, Cambridge, United Kingdom

* acferris@stanford.edu

## Abstract

Cassava brown streak disease (CBSD) is a rapidly spreading viral disease that affects a major food security crop in sub-Saharan Africa. Currently, there are several proposed management interventions to minimize loss in infected fields. Field-scale data comparing the effectiveness of these interventions individually and in combination are limited and expensive to collect. Using a stochastic epidemiological model for the spread and management of CBSD in individual fields, we simulate the effectiveness of a range of management interventions. Specifically we compare the removal of diseased plants by roguing, preferential selection of planting material, deployment of virus-free 'clean seed' and pesticide on crop yield and disease status of individual fields with varying levels of whitefly density crops under low and high disease pressure. We examine management interventions for sustainable production of planting material in clean seed systems and how to improve survey protocols to identify the presence of CBSD in a field or quantify the within-field prevalence of CBSD. We also propose guidelines for practical, actionable recommendations for the deployment of management strategies in regions of sub-Saharan Africa under different disease and whitefly pressure.

**Data Availability Statement:** All of the raw data used to parameterize the model and generate the figures has been previously published. The disease progression time courses and monthly whitefly

## Author summary

Cassava is the second largest source of calories in sub-Saharan Africa and is particularly important for the poorest farmers in the region. Cassava brown streak disease is a viral disease that causes cassava tubers to rot, rendering the roots inedible. Recently, the disease has begun to spread towards major cassava growing regions in West Africa from East Africa, where it continues to cause significant yield losses. Improved approaches for disease control are needed to enable small-holder farmers to prepare for and minimize the impact of the disease when their fields become infected. Using a combination of computational methods and mathematical models enables us to screen a much larger range of potential treatments for their likely effectiveness in managing disease and reducing crop loss than would be possible in conventional field trials, which are expensive and

counts used to parameterize the model are found in Katono et al. [11]. The cassava density data are found in Szyniszewska [23], the whitefly abundance data are found in Jeremiah [13] and Maruthi et al [14], and the CBSD detection data are found in [5,15–22]. The model code is held in a public GitHub repository found at https://github.com/acferris/within-fieldCBSDSpread.

**Funding:** CAG received a grant (RG67082) from the Bill and Melinda Gates Foundation (https://www.gatesfoundation.org). ACF was supported with a fellowship by the Fannie and John Hertz Foundation. The funders had no role in study design, data collection and analysis, decision to publish, or preparation of the manuscript.

**Competing interests:** The authors have declared that no competing interests exist.

logistically difficult to conduct. Our results indicate that regularly planting part of the field with virus-free cassava greatly improves the yield. Removing visibly infected plants and replanting using visibly uninfected plants also improves yield, even when some of these plants may be infected but not yet showing symptoms. We also show how the survey protocol can be optimized to improve estimates of disease severity leading to more effective tailored advice to farmers in regions with different disease pressures.

## Introduction

Cassava is an important food security crop in sub-Saharan Africa. Cassava is typically grown by the poorest households for reliable, subsistence calorific needs, forming the second largest source of calories overall, after maize, in sub-Saharan Africa [1]. Cassava production is currently threatened by cassava brown streak disease (CBSD), which can cause up to a 70% reduction in root yield [2]. The disease is caused by two closely related viruses, cassava brown streak virus (CBSV) and Ugandan CBSV that can be spread by whitefly or by trading or sharing infected cuttings [3]. The disease causes subtle foliar symptoms and brown streaks on the stem as well as root necrosis [4]. Prior to 2004, CBSD was endemic to coastal Eastern Africa and Malawi; however, since 2004 CBSD has undergone a significant range expansion, spreading through Tanzania, Kenya, Uganda, Rwanda Burundi and Zambia into Central Africa, and threatening many new groups of farmers in West Africa [5].

A variety of management strategies have been proposed to limit the impact of CBSD in endemic regions and to reduce the risk of spread into new areas. Roguing plants with foliar symptoms reduces the level of within-field infection although the efficacy of roguing is highly dependent on the ability of farmers to identify disease symptoms, which, in turn, depends on the specific cassava cultivar and levels of farmer training [6,7]. Introducing virus-free planting material (known as 'clean seed') or tolerant varieties with reduced disease symptoms can also be used to improve yield. Tolerance, however, has the disadvantage of maintaining high levels of virus titer in infected plants without displaying symptoms of infection. These require less farmer education but more national-level infrastructure to produce the clean planting material and to coordinate deployment [7,8]. Preferential selection of asymptomatic cuttings is a farmer-level intervention that can potentially reduce the amount of carry-over of infection between successive crops [8]. Alternatively, spraying pesticides or soaking cuttings in pesticide can be used to lower the abundance of the whitefly vector, in order to reduce the amount of within-field spread of the virus.

Although various management techniques have been studied separately, the lack of experimental comparisons within the same study makes it difficult to know when to use one approach over another. Additionally, the effectiveness of these interventions can be highly variable when cultivars and local disease pressures differ. Measuring all of the combinations of management techniques, cultivars, and weather variables is impractical. Epidemiological models provide an alternative tool, allowing us to leverage limited amounts of experimental data in order to make inferences about the effectiveness of different interventions.

Models also help to inform the design and implementation of disease surveillance programmes. Surveillance is an important tool in improving the effectiveness of CBSD control measures, as accurate data on the location of the disease front allows more informed deployment of control measures. Several countries in Eastern Africa have conducted annual CBSD surveys by identifying plants with CBSD foliar symptoms [9]. Because of the lack of access to molecular diagnostic tools in the field and because foliar symptoms can be subtle and vary by

cultivar, assessing the accuracy of field surveys remains a major challenge in monitoring disease levels of CBSD [10]. The problem is further compounded because root symptoms, which allow assessments of yield loss, only appear late in the growing season, when it is no longer possible to identify foliar symptoms. Root surveys are relatively infrequent because of greater time and cost requirements [5]. In this paper we simulate different surveillance practices within- and between-fields in order to identify how changes to current survey protocols could be used to increase accuracy. We also estimate the extent to which current protocols may underestimate the prevalence of CBSD.

We introduce and parameterize a field-level compartmental epidemiological model for the spread of CBSD with explicit vector dynamics. The model is designed to compare the spread of CBSD under conditions of different whitefly densities, local disease pressures, and when different combinations of management tools are used. The model uses experimental time-course data for the spread of the virus and whitefly dynamics [11]. Approximate Bayesian Computation (ABC) is used to estimate the unknown parameters [12]. We apply the model to simulate two agricultural situations: the first applies to individual farmer's fields, where cassava is harvested and replanted yearly and the key output is the total root yield; the second involves clean seed multiplication sites, where cassava plants are ratooned to stimulate the production of secondary stakes. The objective is to maximize the yield of clean (virus-free) stakes, while staying below a given threshold for the proportion of infected stakes. We also use the model to compare different surveillance practices by simulating visual detection of disease under different disease pressures and compare the survey results with the underlying infection state of the field.

Specifically, we address the following questions, focusing on what is realistic and practical for a given stakeholder: for a field in a region with a given vector abundance and disease pressure, what combination and intensity of management strategies are most likely to be effective in reducing the amount and yield loss from CBSD? For individual farmer fields, the effectiveness of management intervention is quantified using yield loss, and for clean seed producers, it is quantified based on the amount of clean planting material produced. For a recently infected field, such as on the epidemic front, we ask how could surveillance be optimised to improve the probability of detecting infection, taking into account the effect of vector abundance and survey variables such as the number of plants surveyed and the accuracy of the diagnostic method (e.g. visual or molecular).

## Results

### Individual farmer's fields

Details of the model structure and simulation are given in the methods. The key parameter values are summarised in Table 1; treatments for managing the disease and vector are summarised in Table 2 and conditions for initial disease levels and whitefly density in Table 3.

Simulations of epidemics within individual farmer's fields were run with annual replanting over a 10 year period in a field with 6000 plants in a 120 x 50 grid (see Table 1). The starting conditions were either low (one infected plant per field) or high disease pressure (25% of plants infected per field) and a range of vector densities (1–20 whitefly that start evenly distributed per top five leaves of each plant). One thousand replicate simulations were run for each management intervention with random starting locations of infected plants within each replicate field. Four different types of management interventions were simulated both individually and in selected combination: clean seed, pesticide coating, roguing, and preferential selection.

Clean seed is implemented by periodically planting a portion of the field with virus-free material with an optional pesticide coating that kills whitefly at the beginning of the season.

**Table 1. Parameter values that were constant in all simulations.**

| Parameters | Parameter value |
|---|---|
| Growing season length | 300 days |
| Plants in field | 6000 |
| Field layout (row x column) | 120 x 50 |
| Spacing within rows | 1m |
| Spacing between rows | 1.5m |
| Plants fully infected | 90 days after infection |
| Symptom delay after infection | 30 days after infection |
| Plants fully symptomatic | 90 days after infection |
| Number of cuttings per plant | 10 |

We compare two levels of clean seed implementation: lower and higher intensity implementation in which, respectively, 10% of the target field is replanted with clean seed every two years or 25% is replanted every three years (Table 2). When roguing is used for disease management, infected plants are removed during two simulated surveys per year. Preferential selection of planting material occurs midway through each season and plants without detected infection are selected to use for replanting the following year. Two efficiencies for detection of infected plants, 33% (low intensity) and 100% (high intensity), are compared for both roguing and preferential selection (Table 2). Three combinations of interventions were simulated: all four interventions (clean seed, pesticide application, roguing and preferential selection) together as a best case scenario, the pair of behavioral interventions (roguing and preferential selection), which could be implemented by an individual farmer with the help of an extension agent, and the pair of infrastructural interventions (clean seed and pesticide coating), which could be implemented if a clean seed system were established. The yield was calculated assuming that all of the plants infected with CBSD at the end of the season achieve 30% of the yield compared to healthy plants [10]; the yields from replicate simulations were averaged. Selected key results from the full analysis are presented here with additional results for reference in the Supplementary Information.

When simulating roguing, there are two distinct outcomes in individual simulations: either all of the infected plants are rogued and the field is free of CBSD, or the field becomes fully infected and there is a negative effect on yield compared with doing nothing because of additional loss of yield from the removal of plants by roguing. Even with perfect roguing accuracy and starting with a single infected plant, fields becomes fully infected in approximately 75% of

**Table 2. Description of management interventions[1] at low and high intensity.**

| Intervention | Description | Low intensity | High intensity |
|---|---|---|---|
| Roguing | Removing symptomatic plants at 3 and 6 mo after planting | 33% accuracy in detecting symptoms of infection | 100% accuracy in detecting symptoms of infection |
| Preferential selection | Selection of asymptomatic stems 6 mo after planting for subsequent planting | 33% accuracy in detecting symptoms of infection | 100% accuracy in detecting symptoms of infection |
| Clean seed | Planting part of the field with virus free plants | 10% of field every three years (L. Good, pers. com.) | 25% of field every two years (L. Good, pers. com.) |
| Pesticide coated clean seed | Planting clean seed coated with pesticide | Lasts for 42 days after planting (J. Colvin, pers. com.) | Lasts for 56 days after planting (J. Colvin, pers. com.) |

[1]Resource requirements are indicated by shading where light grey indicates farmer education and dark grey indicates infrastructure (commercial seed system, pesticide availability).

**Table 3. Summary of disease pressure and whitefly settings for simulations with low or high disease pressure.**

| Epidemiological driver | Value |
|---|---|
| Low disease pressure | 1 infected plant per field |
| High disease pressure | 25% infected plants per field |
| Whitefly | {1, 5, 10, 15, 20} whitefly per top 5 leaves of plant |

the simulations because the incubation period results in a large number of infected plants that have yet to develop visible symptoms early enough during the roguing period (Fig 1b). High accuracy roguing, by which we mean infected plants are always detected if surveyed, is more likely to eliminate CBSD from the field but when elimination fails, continued roguing would eventually lead to the nonsensical result of removing all plants leading to complete loss of yield (Fig 1b and 1d). Overall, roguing only increases the average yield when there is very low whitefly abundance and there is either low disease pressure or high disease pressure with high accuracy roguing (S1 Fig).

Our results indicate that the deployment of clean seed was more successful in reducing potential crop loss from CBSD than other separately applied management interventions (roguing of symptomatic plants and preferential selection of planting material) (S1 Fig). The yield advantage of using clean seed decreased with increasing vector densities, necessitating the introduction of high intensity clean seed for effective control even at five whitefly on upper leaves per plant (Fig 2b and 2c). The number of subsequent years where there was a yield improvement from clean seed also decreased with higher whitefly abundance and starting infection levels. The effect of clean seed on yield is also much larger when planted as a single

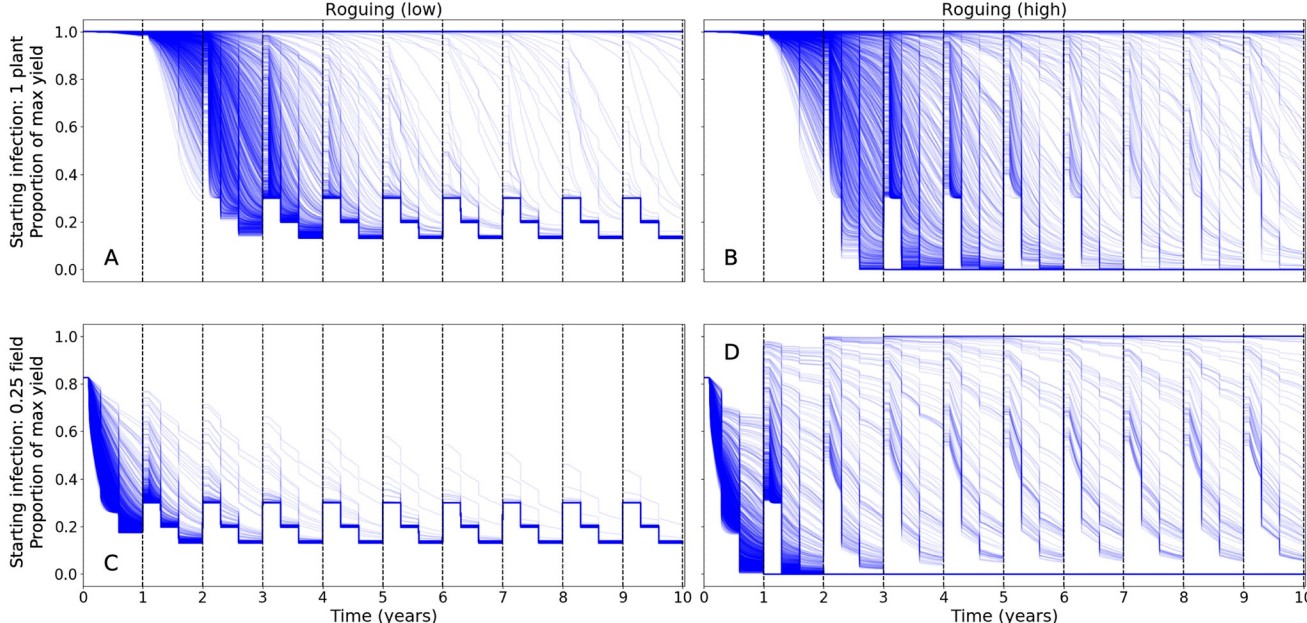

**Fig 1. Effect of high and low roguing accuracy in detecting and removing infected plants on potential cassava yield loss across multiple years.** One thousand independent simulation runs are shown for each treatment combination of roguing accuracy x initial CBSD density and initial density of five whitefly per top five leaves per plant. The initial decrease in average yield 30 days into the growing season corresponds to when it is possible for new plants to be infected and the two drops in yield at 90 and 180 days correspond to roguing. (A,B) the initial CBSD infection is a single plant; (C,D) initial CBSD infection is 25% of plants per field. (A,C) have a low roguing accuracy; (B,D) have high roguing accuracy (see Table 2). Darker intensity lines indicate multiple simulation lines overlapping. For each sub-plot, the x-axis is the time since the epidemic began, with each season demarcated by vertical dashed lines: the y-axis is the potential yield during that season relative to a healthy field if no additional plants are infected or rogued (see text for details). The final effective yield for a season is given by the value of yield at the end of a season.

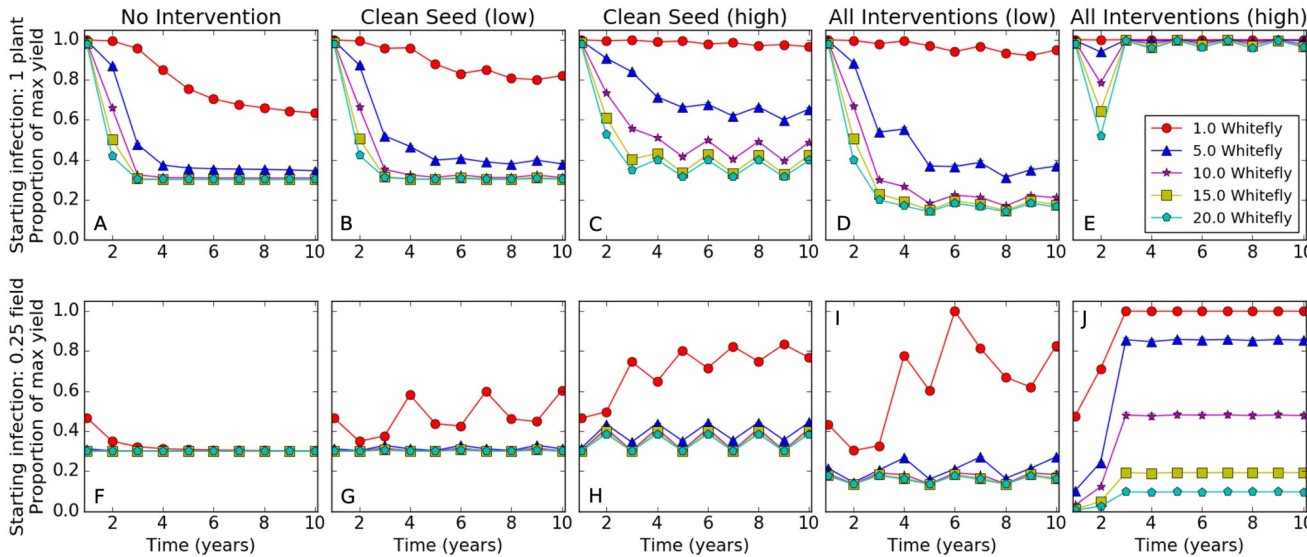

**Fig 2. The average yield of a field with different management interventions.** For each plot, the x-axis is the number of seasons since the start of the epidemic, and the y-axis is the average yield over an ensemble of 1000 epidemics for each management intervention. (A,B,C,D,E) have a starting infection of a single plant while (F,G,H,I,J) start with 25% of the field infected. (A,F) have no interventions, (B,G) use low intensity clean seed, (C,H) use high intensity clean seed, (D,I) use all interventions at low intensity, and (E,J) use all interventions at high intensity (see Table 2).

block in the corner of a field as opposed to being randomly distributed throughout the field. The difference was especially pronounced when there were five whitefly per plant and a low starting infection. High intensity clean seed improved yield by 13% when randomly planted compared with 89% when block planted. The differences were respectively 48% and 0% with a high starting infection. (cf Fig 2 for blocked and S2 Fig for random introduction of clean seed). In the simulations, randomly distributed cuttings are quickly infected by neighboring infected plants, whereas it takes longer for the infection to spread completely through a large block of uninfected clean seed.

Almost all the management interventions increased yield to some extent but only at the lowest vector density; the exception being low intensity roguing in endemic fields (cf S1 Fig). Preferential selection of planting material on its own had a minimal impact on yield, and in combination with roguing decreased yield except for the lowest vector density (S1 Fig). When combining the high intensity interventions, the fields were rendered free of CBSD within three years with low starting CBSD infection (Fig 2e).

## Regional management recommendations

We propose the following field level recommendations for management interventions to use in different regions of sub-Saharan Africa based on the likely disease pressure, crop density, and whitefly abundance. Exploratory simulations showed that epidemic predictions behaved differently at low and high disease pressures according to whitefly abundance. Survey data were used to classify locations with respect to (very low to high) whitefly abundance [13,14]. High disease pressure areas were defined as locations within 500km radius of a known CBSD positive survey point or in regions with high cassava density [5,15–23]. Therefore, parts of Nigeria have been classified as having high disease pressure, even though CBSD has not been reported in West Africa, based on the assumption that if CBSD reaches the region the disease would spread more quickly in places with high cassava density. Other regions were classified as having low disease pressure.

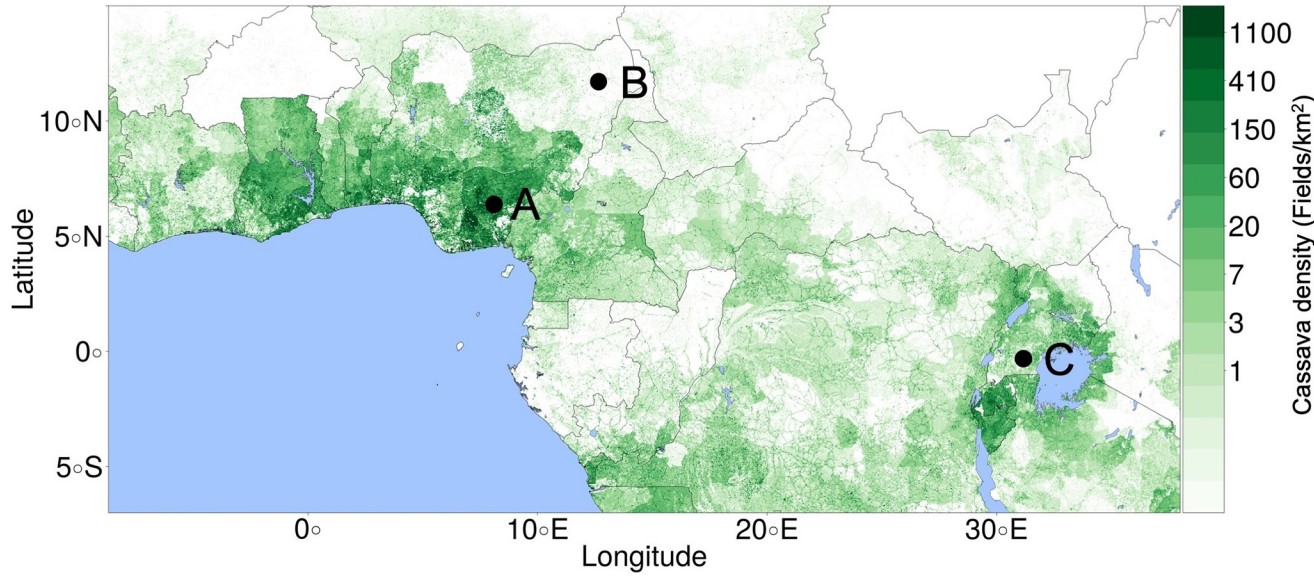

| Location | Whitefly abundance | Local Disease pressure | Recommendations in Order of Effectiveness |
|---|---|---|---|
| A | Very low | High* | 1. Clean seed<br>2. Clean seed with pesticide coating<br>3. Roguing<br>4. Preferential selection |
| B | Moderate | Low* | 1. Clean seed<br>2. Clean seed with pesticide coating |
| C | High | High | 1. High intensity clean seed<br>2. High intensity clean seed with pesticide coating |

**Fig 3. Recommended interventions for improving yield in CBSD infected fields in selected regions based on whitefly abundance, host density, and disease pressure.** Asterisks indicate regions that do not currently have CBSD and what the disease pressure is likely to be if CBSD were present. Recommendations are listed in order of decreasing impact on yield. For locations A and B, low intensity interventions are effective but high intensity would work better; however, for location C only high intensity is likely to be effective (see Table 2).

In most of West Africa there is low whitefly abundance, which increases the effectiveness of all the management interventions, particularly roguing and preferential selection (Fig 3 and S1 Fig). In regions with moderate whitefly but low disease pressure, roguing and preferential selection become ineffective, but clean seed and pesticide coating are somewhat effective at low intensity and more so at high intensities (Fig 2 and S1 Fig). With higher disease pressure and whitefly abundance, high intensity clean seed and pesticide coating are needed to improve yield (Fig 2 and S1 Fig).

## Multiplication in clean seed nurseries

Given evidence for the effectiveness of clean seed as a management intervention, we also simulated how to use management interventions effectively to increase the output and sustainability of clean seed multiplication. Unlike within farmer's fields, in these simulations, nursery fields

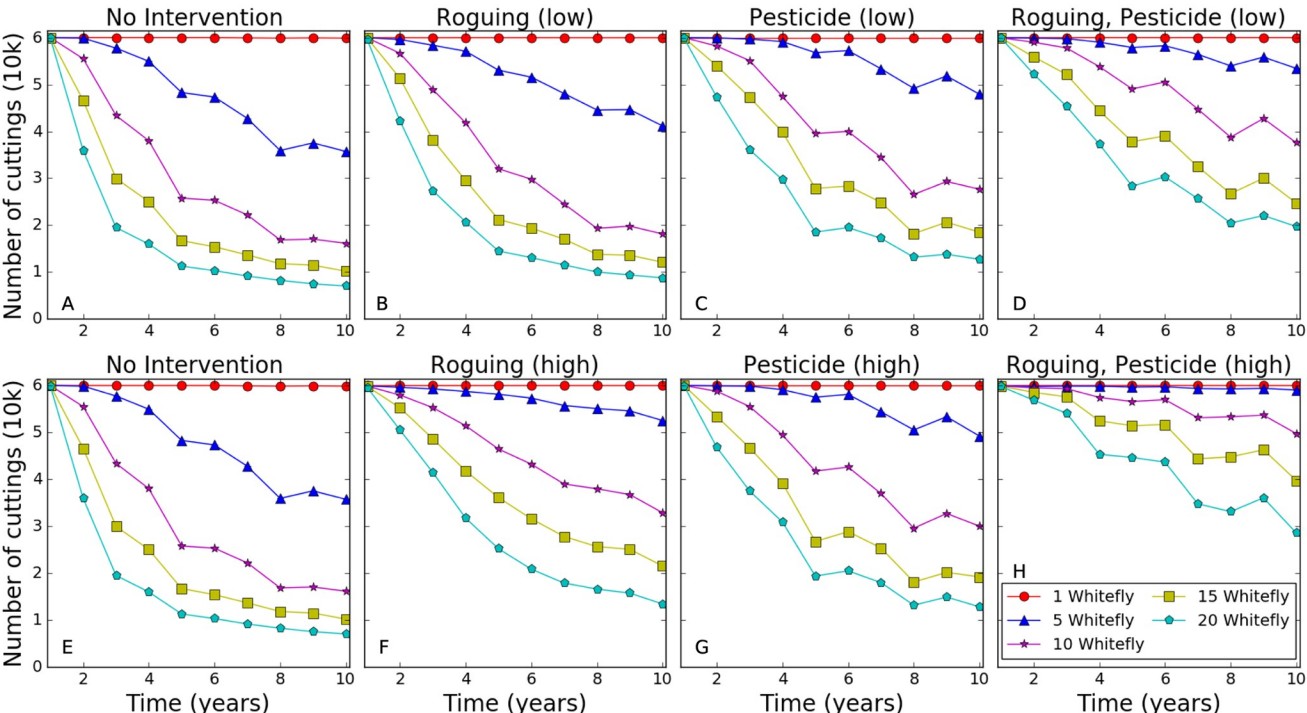

**Fig 4. Average number of cuttings produced by fields for clean seed production with different management interventions.** For each plot, the x-axis is the number of seasons, and the y-axis is the average number of cuttings produced in a field over an ensemble of 1000 simulations. (A,E) are the no intervention baseline results, (B,C,D) have a low intervention intensity, and (F,G,H) have a high intervention intensity (see Table 2). All simulations relate to low initial density of one infected plant per field.

for the production of clean planting material are ratooned every three years. The specific simulation settings were based on the local level of clean seed multiplication for nurseries from Tanzania's proposed clean seed system for which fields are used for supply of planting material up to a threshold level of 10% within-field infected plants [24] (L. Good, pers. com.).

Roguing and pesticide coating on newly planted clean seed each increased the number of clean cuttings generated compared with no interventions under low whitefly abundance (Fig 4). With high whitefly abundance, however, using pesticide resulted in a small improvement, and there is no improvement with low accuracy roguing. Increasing the roguing accuracy from low to high also led to a large improvement in the average number of cuttings generated, suggesting that cassava varieties where it is easier to detect foliar symptoms should be prioritized for use in clean seed systems to improve sustainability over time (Fig 4b and 4f). There was also a strong synergistic effect when combining roguing and pesticide treatment for all levels of whitefly abundance (Fig 4).

The clean seed requirements for a region varies depending on the frequency that farmers introduce clean seed to their fields and the percentage of the field that is to be replanted. A single clean seed multiplication field could support up to 1800 farmers using low intensity clean seed (replanting 10% of a field every three years) or up to 750 farmers using high intensity clean seed (replanting 25% of a field every two years), assuming that none of the plants in the field become infected (Table 2). This means that a change of 20,000 cuttings produced, for example by using low intensity roguing and pesticide in an area with moderately abundant whitefly, would translate to a clean seed producer being able to support an additional 100 farmers using low intensity clean seed or 27 using high intensity.

## Surveillance

We used the simulation model to compare the effectiveness of a range of within-field surveillance practices in successfully detecting CBSD disease in infected fields and in estimating the level of within-field disease incidence. The basic survey technique is described in the methods section and follows the protocol described by Sseruwagi *et al.* [9]. We simulated the effects of different combinations of sampling intensity (numbers of plants assessed for disease) and accuracy (correctly assigning an infected plant as diseased) on the probability of detecting disease. In practice survey accuracy can vary depending on the type of diagnostic test used to assess disease (e.g. root symptoms, foliar symptoms, or molecular diagnostics). Accuracy also depends on the severity of foliar symptoms in the local cultivars, and on surveyor expertise.

When 5% of a field is infected with CBSD at the beginning of a growing season, surveying at least 30 plants per field results in a high probability of detecting infection. Conducting foliar surveys later in the growing season increases the likelihood of detecting disease, regardless of survey accuracy and whitefly abundance (Fig 5). Conducting surveys later in the season greatly improves the probability of detecting infection in all cases except with a very low whitefly density (one whitefly per plant) because CBSD within-field abundance increases enough over the course of the growing season to improve the probability of detecting the infection (Fig 5). The probability of detecting CBSD is very low when there is only one infected cutting in the field regardless of survey accuracy, survey timing, or the number of plants surveyed (S3 Fig). Similarly, when greater than 20% of the field are infected, the survey will almost always detect CBSD irrespective of the survey parameters (S3 Fig).

Estimating the within-field prevalence of CBSD in fields where the disease is established is highly dependent on survey accuracy, but not on the number of plants surveyed in the field (Fig 6 and S4 Fig). However, when the disease is present at a very low level in the field, the survey accuracy is low regardless of the survey protocol (S4 Fig). Conducting foliar surveying later in the season has a moderate effect on the proportional error of the estimates for

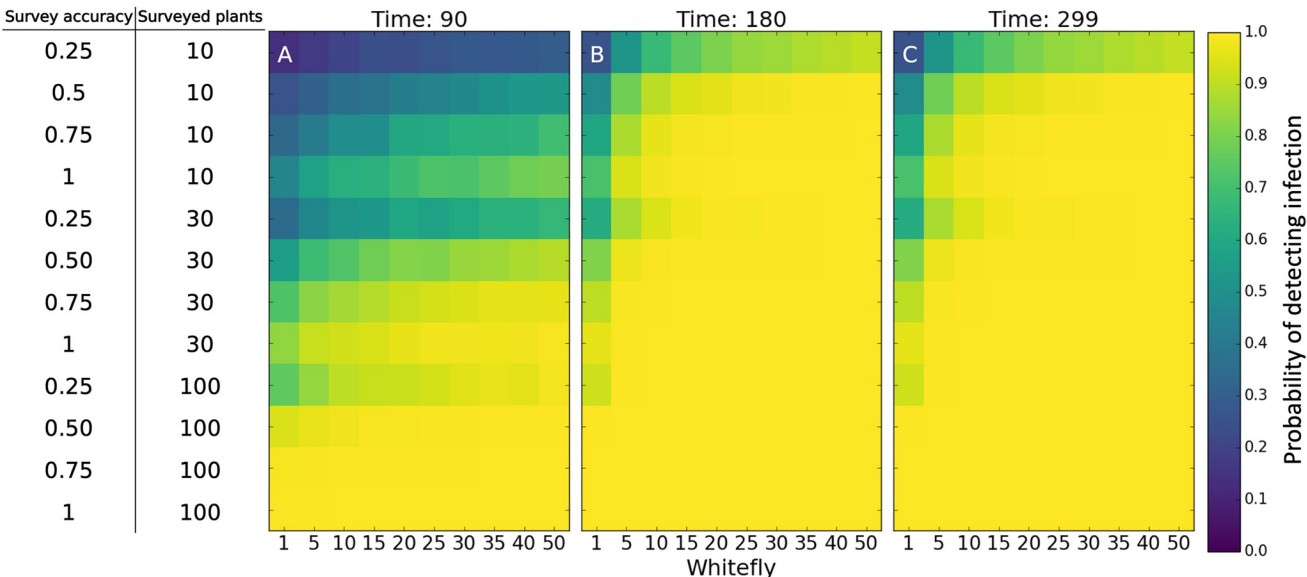

**Fig 5. The probability of detecting infection in a field with an initial within-field incidence of 5%.** For each plot, the x-axis is the average number of whitefly per top five leaves on the plant, and the y-axis is a different combination of survey variables (surveyor accuracy and number of plants surveyed). Simulated surveys were conducted at (A) 90 days after planting; (B) 180 days after planting; (C) 299 days after planting.

 Computational models to improve surveillance for CBSD and minimize yield loss

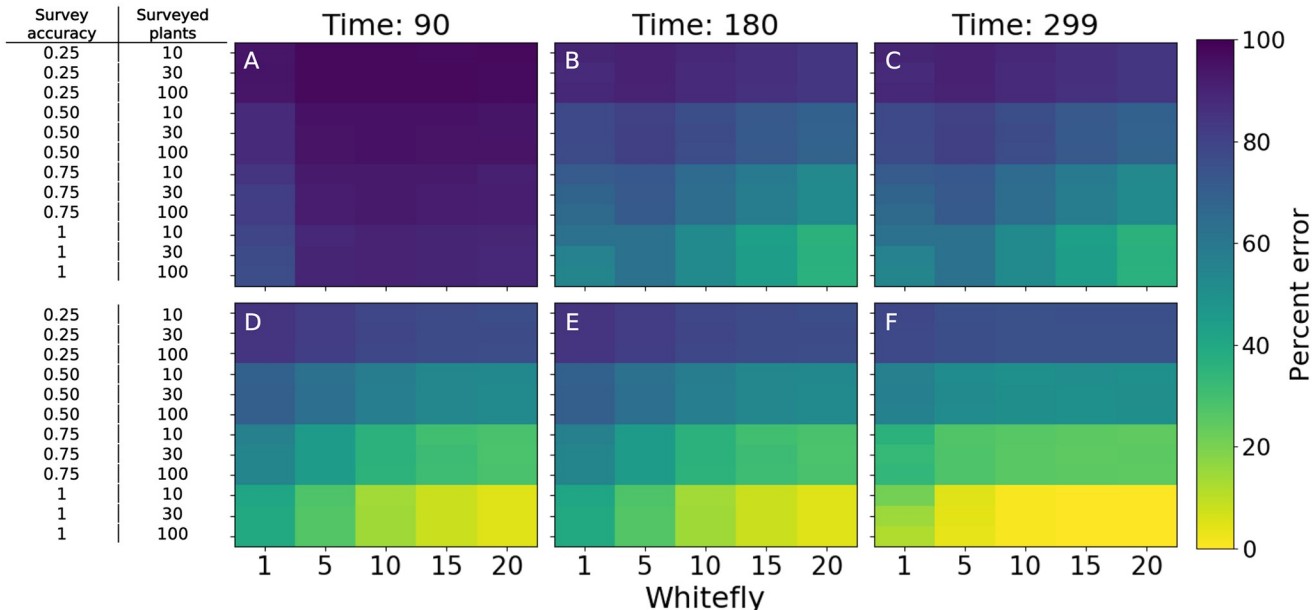

**Fig 6. The percent error of within-field CBSD prevalence estimated from a survey compared with the true prevalence at the end of the season.** For each plot, the x-axis is the average number of whitefly per top five leaves on the plant, and the y-axis is a different combination of survey variables (surveyor accuracy and number of plants surveyed). (A,D) have surveys conducted 90 days after planting, (B,E) have surveys conducted 180 days after planting, and (C,F) have surveys conducted 299 days after planting. (A,B,C) have a starting infection of 5% and (D,E,F) have a starting infection of 20%.

simulated epidemics, but only for fields infected with low levels of CBSD (Fig 6). This implies that replacing standard foliar surveys with molecular diagnostics or root surveys at the end of the season (each of which is associated with greater accuracy) could increase survey accuracy, but in general, estimating the prevalence of CBSD in fields with an initial infection of less than 20% will have a high proportional error.

## Discussion

Our study demonstrates that roguing, preferential selection of asymptomatic planting material, and clean seed are all individually effective at reducing epidemic spread and improving yield in fields within areas of low disease pressure and low whitefly abundance. In regions with high whitefly abundance or high disease pressure, the use of clean seed is essential to improve average yield. Clean seed nurseries were able to remain below a critical CBSD infection limit of 10% for several periods in regions with low to moderate whitefly abundance, particularly when roguing or pesticide coating are also used. Our results also indicate that surveillance to detect CBSD when it is present at low abundance (5%) can be optimized by conducting surveys later in the growing season and surveying at least 30 plants per field.

### Individual farmer's fields & clean seed multiplication

We simulated two different types of management interventions, those that depend on farmer education (roguing and preferential selection), and those that rely on external inputs (clean seed and pesticide coating). Current guidelines for cassava recommend selecting healthy plants mid-season and roguing diseased plants [25,26]. However, previous studies show that farmers in regions with CBSD are unaware of the disease and that training from extension agents is required for farmers to be able to recognize infected plants in their fields [8,27]. Based on our simulation results, preferential selection of asymptomatic planting material does not decrease

the average yields, but only results in a large yield increase in regions that have very low levels of whitefly and for plants with easily detectable foliar symptoms (Fig 3 and S1 Fig). Assuming that the practice increases accuracy of detection, training farmers to select plants at the end of the season based on root symptoms as opposed to selecting plants midseason using foliar symptoms [25] would be expected to increase the number of regions that can benefit from preferential selection.

The effectiveness of roguing has been tested previously in field trials in conjunction with planting clean seed: the practice was beneficial in one trial location and had no effect on yield in the other location [7]. Our simulation data suggest that when there is moderate or high whitefly abundance, roguing decreases yield, even with perfect accuracy (S1 Fig). However, without roguing, these fields act as a viral reservoir that is more likely to spread CBSD to neighboring fields, so roguing may confer benefits at a regional scale even if it is likely to reduce the yield for an individual farmer. Instead of only roguing infected plants, shifting to a strategy of also removing plants within a set radius [28] of any plants that are found to be symptomatic may increase the efficacy when there is higher whitefly abundance. The optimal radius for roguing [cf 28] would likely be highly dependent on disease pressure and whitefly population levels, and in the interest of brevity this has been left for future work.

The simulations suggest that using high intensity clean seed (replanting a quarter of the field with clean seed every other year) always improves average yield, and that in regions with very low abundance of whitefly, less frequent input is also beneficial (Fig 2). Legg *et al.* [7] also reported yield improvements in field trials from introducing clean seed in a variety of locations with different whitefly abundance and disease pressure. The simulation results also suggest that sustainably multiplying uninfected cuttings to use as input clean seed for farmers is feasible using the proposed Tanzanian clean seed system (replanting a third of the field every year with a within-field infection limit of 10%), as long as the multiplication is done in areas with low whitefly and roguing or pesticide is used (Fig 4) [24] (L. Good, pers. com.). These conclusions support previous simulation results by McQuaid *et al.* [6] that clean seed multiplication is sustainable in regions with low disease pressure and whitefly density, particularly when there is effective roguing. However, it is difficult to make more detailed comparisons because McQuaid *et al.* [6] used a very different set of parameters based on cassava mosaic disease and assumed that roguing occurred at least monthly with at least 50% accuracy.

Another potential benefit of clean seed is distributing new cultivars with other beneficial characteristics. Varieties with obvious foliar symptoms can increase the ease of detecting infected plants and hence the effectiveness of roguing and preferential selection. Cultivars with early bulking roots can be harvested earlier in the season, minimizing the amount of root necrosis [8,29]. Currently there are no resistant varieties where the plant is able to eliminate the virus after being infected; however, tolerant varieties of CBSV have been identified that have reduced root necrosis when infected with CBSV [30,31]. Some of these tolerant varieties have reduced foliar symptoms that may or may not be associated with reduced root symptoms. Such varieties create a reservoir of infection. The absence of foliar symptoms reduces the efficiency of roguing and preferential selection as farmers are unaware that their field is infested with CBSV.

Whitefly abundance plays a key role in determining which interventions were beneficial and the magnitude of their effectiveness. More management interventions are effective and have greater impact under conditions of lower whitefly abundance There is a higher overall average yield in fields at the disease front with five or fewer whitefly per top five leaves, even in the absence of management interventions. Although not directly simulated in the paper, pursuing approaches that reduce whitefly abundance, for example by use of pesticide would likely be an effective strategy for managing CBSD.

## Surveillance

The model results suggest that when seeking to detect the presence or absence of CBSD in a field, the survey protocol is particularly important at relatively low within-field CBSD abundance. With higher vector abundance, there is an advantage to surveying later in the season because there is enough within-field CBSD spread during the season to meaningfully increase the probability of detecting an infected plant (Fig 5). Surveying at least 30 plants per field with a survey accuracy of at least 50% is important, but further increasing the survey accuracy has a minimal impact (Fig 5). Using molecular diagnostics in conjunction with visual symptom observation or conducting surveys using root symptoms instead of foliar symptoms (to improve accuracy of detection) could be a way to extend the window of time for doing surveys that have a high probability of detecting an infection. In practice, most fields are probably not identified by surveys until closer to 5% of the plants are infected (S1 Fig). This threshold for detecting infected fields suggests that there is a lag between the disease front and detection of infected fields, and that control efforts should be expanded beyond the region where infected plants have been detected.

Accurately estimating the within-field prevalence of CBSD with surveys is a more difficult problem. When the within-field prevalence is 5% or less, the percent error is high regardless of the survey variables used. Even for fields with a starting infection of 20% or 50%, the vector prevalence, survey timing, the number of plants surveyed all have a small effect on percent error. Survey accuracy is by far the most important of the variables under surveillance control, suggesting that a survey protocol that analyzes ten plants using molecular diagnostics would be a reasonably effective approach (Fig 6). Due to the large differences in optimal survey protocols for estimating whether a field is infected and for estimating with-in field prevalence, it is not feasible to meet both objectives using a single survey data protocol.

In conclusion, we find that for much of East and West Africa, preferential selection, roguing, and clean seed can all improve yield, while clean seed is required to improve yield in areas with high disease pressure or whitefly abundance. Our findings also demonstrate that clean seed can be generated sustainably in moderate to low whitefly areas, particularly when roguing or pesticide coating is used. Surveys to detect the presence of CBSD can be highly accurate even close to the epidemic front, allowing for effective targeting of management interventions. However, accurately quantifying the within-field prevalence requires a higher survey accuracy than is unlikely to be achieved using foliar symptoms alone.

## Methods

### Modelling approach

Plants become infected in a field either through viruliferous whitefly or planting already infected cuttings. In the model, plants are initially cryptically infected with a 30 day lag before symptoms start to become visible at which point symptoms increase linearly until 90 days after infection. This reflects results from Mware et al. [15] that foliar symptoms from cassava brown streak virus develop 26–60 days after exposure to infected whitefly (Table 1, Fig 7A). We also assume that immediately after a plant becomes infected there is zero chance for emigrating whitefly to carry the virus, and that the infectiousness of a plant increases linearly until it becomes maximally infectious after 90 days, reflecting the experimental results of Rwegasira and Rey [32] (Table 1, Fig 7A). The field size for all of the simulations was 6000 plants, which is typical of smallholder cassava fields, and is large relative to the dispersal kernel, Field dimensions and other critical parameters are summarized in Table 1.

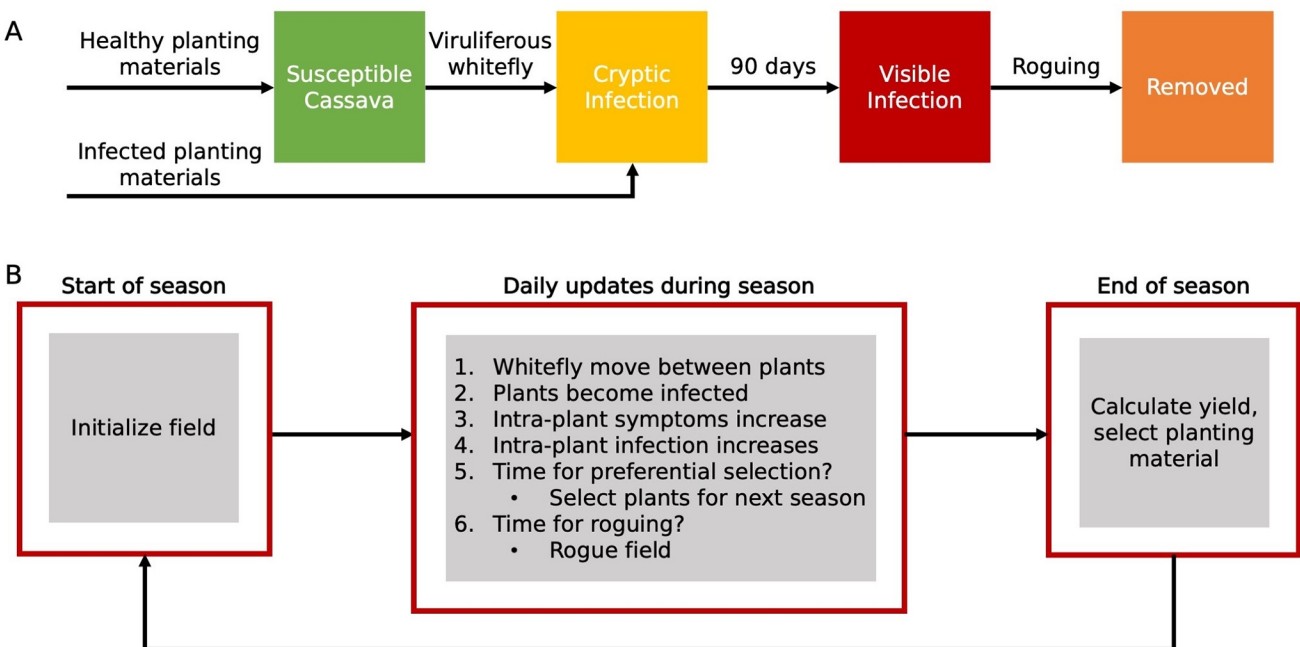

**Fig 7. Overview of model structure.** (A) shows the infection process for hosts progressing from susceptible and uninfected to cryptically infected. (B) shows the model over the course of a single growing season. At the start of the simulation, the field is initialized with the location of infected plants and whitefly. Roguing and preferential selection occur at set intervals during the season whereas whitefly movement, new infections, and intra-plant symptom and infection increases occur daily.

We model the spread of CBSD within a field using a spatially-explicit, individual-based stochastic model that explicitly models individual cassava plants and whitefly in a field. Each plant holds a population of whitefly. Whitefly movements between plants are updated in daily time steps, and each whitefly has a chance to fly to another plant every day (Fig 7B). The destination plant for a whitefly movement is chosen by a weighted random sample with each potential destination weighted by an exponential kernel. As whitefly only transiently carry the CBSV virus, a check is made upon whitefly emigration from an infectious plant to see whether the insect is carrying the virus. Reflecting results from Maruthi et al. [14] showing that with intermittent feeding CBSV is retained for less than 24hrs, we assume that whitefly only retain the virus for the first feeding after dispersal because in the model whitefly only move between plants once a day. When parameterizing the model, the maximum transmission probability was set at 12.5% reflecting experimental results [15,33,34].

At the beginning of a growing season, replanting of a field is simulated by selecting cuttings from the previous year, and the initial level of within-field infection is the average of the infection of the selected plants and any clean seed (Fig 7B). On each day during the growing season, whitefly move between plants, new plants become infected by whitefly, and the symptom severity and infectiousness increase. Roguing infected plants and preferential selection of plants to use in the subsequent growing season can also occur midway through the growing season (Fig 7B). At the end of the season, the yield of the field is calculated using a simple relationship between infection and yield assuming that a CBSD infected plant has 30% of the yield compared with a healthy plant [10] (Fig 7B). Annotated model code with example input files are available in GitHub repository within-fieldCBSDSpread (https://github.com/acferris/within-fieldCBSDSpread).

## Simulating clean seed nurseries

When simulating clean seed nurseries, ratooning is used instead of replanting at the end of the growing season. To simulate ratooning, the field is divided into sections, which are replanted every three years using clean seed, and any plants removed by roguing are not replaced until that section of the field is replanted. This protocol mimics the guidelines given to cassava seed entrepreneurs in the clean seed system being developed by Tanzanian scientists in partnership with the Mennonite Economic Development Associates [24]. Fields are removed entirely when more than 10% of the plants are infected at the end of season (L. Good, pers. comm.). The effectiveness of different management techniques was quantified by calculating the number of cuttings produced by a field. All live plants produced ten cuttings and rogued plants produced zero cuttings (see Table 1). If the average CBSD infection in the cuttings was above the cutoff value of 10%, the field produced no cuttings for the rest of the simulation.

## Modelling management interventions

Roguing is modelled by simulating a survey of all plants in a field, then removing any plants detected as symptomatic three and six months into the growing season (Table 2). For preferential selection, plants are surveyed 180 days after planting, and a subset of plants without visible symptoms is pre-selected as planting material for the following year (Table 2). A scale factor can be applied to the plant symptom value to mimic imperfect surveillance or cultivars with less obvious foliar symptoms.

The model allows for flexibility in selecting planting material: for example, clean seed (i.e. uninfected plants) can be added to specific regions of a field, and preferential selection starts by trying to choose plants from the clean seed region of the field. Clean seed can be combined with a pesticide coating, which is modelled by killing any whitefly that land on the plant before the pesticide loses effectiveness midway through the season (J. Colvin, pers. com.). To maintain a constant number of whitefly per field, a new whitefly is added to a random plant after one is killed by pesticide and more whitefly are added to the plants when the pesticide efficacy declines to bring the plants up to the field average. The differences between the low and high intensity versions of all four interventions are summarised in Table 2, and the differences in initial conditions for the simulations are recorded in Table 3. The efficacy of the different management techniques were compared by estimating the average yield of a field over time, using a simple relationship between infection and yield assuming that a CBSD infected plant has 30% of the yield compared with a healthy plant [10].

## Modelling surveys

To simulate surveys, we assume that the probability of detecting an infected plant is proportional to the extent of symptoms for that plant using the expression: (*proportion of max symptoms*) * (*survey accuracy*). Varieties with less severe foliar and/or root symptoms are modelled by using a survey accuracy value less than 1. Surveys select only a subset of N randomly selected plants approximately uniformly spaced along two diagonal transects of a field [9].

## Parameter estimation

Approximate Bayesian computation (ABC) methods [12] were used to estimate the epidemiological parameters from the Kamuli dataset [11]. The data from Katono *et al.* [11] was collected at two different locations in Uganda, Wakiso (high CBSD prevalence) and Kamuli (medium CBSD prevalence). At each test site, the CBSD susceptible TME204 cassava variety was grown in 100 plant blocks in a 10 x10 square lattice with 1m between rows of plants. There were four

replicate blocks with 2m between each block. Researchers visually surveyed the plants for foliar symptoms and counted the number of whitefly monthly from 1 to 12 months after planting. The data also included whitefly prevalence data from three different field sites, which all had different whitefly dynamics. Because there are not enough data from other sources about whitefly dynamics within a growth season, for simplicity whitefly abundance was assumed to be the average of the recorded values at a location with no seasonal dynamics.

Three parameters were estimated using the Kamuli dataset [11]: (i) probability of a whitefly leaving a plant per day; (ii) probability of a whitefly infecting a new plant; (iii) dispersal kernel scale factor for an exponential kernel is given by: $e^{-(kernel\ scale\ *\ distance\ between\ plants)}$ We used the summary statistic: $\sum \frac{(experimental\ value - simulated\ value)^2}{experimental\ value}$ (analogous to the $\chi 2$ statistic) and explored a range of tolerance values, selecting 15 as a robust measure (S5 Fig).

To validate that the posterior parameter values were robust to different field conditions, we used the parameters obtained from fitting to the Kamuli dataset and ran simulations using the field conditions from the training dataset, Kamuli, and the validation dataset, Wakiso. The percentage of simulation runs with a summary statistic value below the cutoff were higher for the validation dataset than the training dataset, suggesting that the parameters generalize beyond a single geographic location (S6 Fig).

## Supporting information

**S1 Fig. The difference in average yield between fields with management interventions and a field with no management interventions.** (A,B,C,D,E) start with one infected plant and use low intensity interventions, (F,G,H,I,J) start with one infected plant and use high intensity interventions, (K,L,M,N,O) start with a quarter of the field infected and use low intensity interventions, and (P,Q,R,S,T) start with a quarter of the field infected and use high intensity interventions. For each subplot, the x-axis is the number of seasons since the start of the epidemic. (TIF)

**S2 Fig. The average yield of a field using randomly planted clean seed.** For each plot, the x-axis is the number of seasons since the start of the epidemic, and the y-axis is the average yield over an ensemble of 1000 epidemics for each management intervention. (A,B,C) have a starting infection of a single plant while (D,E,F) start with 25% of the field infected. (A,D) have no interventions, (B,E) use low intensity randomly planted clean seed, and (C,F) use high intensity randomly planted clean seed.
(TIF)

**S3 Fig. The probability of detecting infection in a field with an initial within-field incidence of 1 infected plant.** For each plot, the x-axis is the average number of whitefly per top five leaves on the plant, and the y-axis is a different combination of survey parameters (surveyor accuracy and number of plants surveyed). (A,B,C) have a starting infection of a single plant while (D,E,F) start with 20% of the field infected. The surveys were conducted at (A,D) 90 days after planting; (B,E) 180 days after planting; (C,F) 299 days after planting.
(TIF)

**S4 Fig. The percent error of within-field CBSD prevalence estimated from a survey compared with the true prevalence at the end of the season.** For each plot, the x-axis is the average number of whitefly per top five leaves on the plant, and the y-axis is a different combination of survey variables (surveyor accuracy and number of plants surveyed). (A,D) have surveys conducted 90 days after planting, (B,E) have surveys conducted 180 days after planting, and (C,F) have surveys conducted 299 days after planting. (A,B,C) have a starting infection of

one plant and (D,E,F) have a starting infection of 50%.
(TIF)

**S5 Fig. Sample simulation results and posterior probabilities for the parameters used in the simulations.** (A) shows experimental data from Katono *et al.* in cyan with an example of simulated data with an accepted set of parameters shown as a red line. Double headed arrows show the distance between the experimental and simulated data. (B,C,D) show marginal posterior probabilities for the parameter values.
(TIF)

**S6 Fig. The distribution of summary statistic scores from 5000 simulation runs using the accepted parameters and whitefly abundance from (A) Kamuli or (B) Wakiso.** The red line indicates a summary statistic score of 15. Thirty one percent of the Wakiso simulations are below the cutoff value of 15 and 15% of the Kamuli simulations are below the cutoff.
(TIF)

**S1 Movie. An example time course for a field with no management interventions and an average of 15 whitefly/top five leaves.** Uninfected plants are shown in navy, infected plants are shown in yellow.
(MP4)

## Acknowledgments

The authors gratefully acknowledge discussion with Professor John Colvin on whitefly behaviour and with Dr Lauren Good on strategies for clean seed deployment.

## Author Contributions

**Conceptualization:** Alex C. Ferris, Richard O. J. H. Stutt, David Godding, Christopher A. Gilligan.

**Data curation:** Alex C. Ferris.

**Funding acquisition:** Christopher A. Gilligan.

**Investigation:** Alex C. Ferris.

**Software:** Alex C. Ferris, David Godding.

**Supervision:** Richard O. J. H. Stutt, Christopher A. Gilligan.

**Validation:** Alex C. Ferris, David Godding.

**Visualization:** Alex C. Ferris.

**Writing – original draft:** Alex C. Ferris.

**Writing – review & editing:** Alex C. Ferris, Richard O. J. H. Stutt, Christopher A. Gilligan.

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
