## [Decision Letter · Decision Letter 0]

13 Jan 2020

Dear Dr Ferris,

Thank you very much for submitting your manuscript, 'Computational models to improve surveillance for cassava brown streak disease and minimize yield loss.', to PLOS Computational Biology. As with all papers submitted to the journal, yours was fully evaluated by the PLOS Computational Biology editorial team, and in this case, by independent peer reviewers. The reviewers appreciated the attention to an important topic but identified some aspects of the manuscript that should be improved.

We would therefore like to ask you to modify the manuscript according to the review recommendations before we can consider your manuscript for acceptance. Your revisions should address the specific points made by each reviewer and we encourage you to respond to particular issues Please note while forming your response, if your article is accepted, you may have the opportunity to make the peer review history publicly available. The record will include editor decision letters (with reviews) and your responses to reviewer comments. If eligible, we will contact you to opt in or out.

- Supporting Information uploaded as separate files, titled 'Dataset', 'Figure', 'Table', 'Text', 'Protocol', 'Audio', or 'Video'.

We hope to receive your revised manuscript within the next 30 days. If you anticipate any delay in its return, we ask that you let us know the expected resubmission date by email at ploscompbiol@plos.org.

Sincerely,

Konstantin B. Blyuss

Guest Editor

PLOS Computational Biology

Virginia Pitzer

Deputy Editor

PLOS Computational Biology

[LINK]

Reviewer's Responses to Questions

**Comments to the Authors:**

Reviewer #1: I believe that this manuscript should be published. I have I believe the topic is relevant and an important investigation of intervention efficacy in a sustainable manner. I believe that this manuscript should be published, however, I have some points I would like to bring the authors attention to:

line 241-247: I found that stating "Increasing the roguing accuracy from low to high also led to a large improvement in the average number of cuttings generated.." slightly misleading. Figure 4 shows modest increases, and the trends in both look to be the same. Looking at the scale, which reflects that increases are by 10k, there isn't a benchmark for me to know if that is indeed a large increase, e.g. if 6000 cuttings creates 1 full additional field, the gains seem modest, unless I have missed something. I would recommend that presenting the information with an appropriate relative scale.

line 266-269: I find that the sentence does not read well and could be better written.

line 297: "...combination of survey variables (surveyor accuracy number of plants surveyed)..." It seems to me there is a missing word within the parentheses.

line 494: is this a negative exponential kernel? The kernel is not described, unless I have missed this. I would like to see this.

For the authors interest in increasing surveillance I would like to bring their attention to the app called Plantix if they have not heard of this before.

https://plantix.net/en/

It has been a very successful diagnostic framework within India for pests and diseases of agricrops. Attempts are under way to use data collected from Plantix to help with detection and prevention of crop losses from pests and diseases. I feel that discussions with the Plantix team would benefit future work for CBSD in West Africa.

Reviewer #2: Dear authors, please find attached two documents of my review. A word document with general comments and an annotated PDF version of the submitted manuscript with detailed comments.

**Have all data underlying the figures and results presented in the manuscript been provided?**

Reviewer #1: No: I could not find a link to the data, perhaps I have missed this?

Reviewer #2: No: Would the authors consider submitting an annotated code of the model?

PLOS authors have the option to publish the peer review history of their article (what does this mean?). If published, this will include your full peer review and any attached files.

Reviewer #1: Yes: Vincent A. Keenan

Reviewer #2: No

---

## [Editor Report · Decision Letter 1]

24 Mar 2020

Dear Mx. Ferris,

We are pleased to inform you that your manuscript 'Computational models to improve surveillance for cassava brown streak disease and minimize yield loss.' has been provisionally accepted for publication in PLOS Computational Biology.

Best regards,

Konstantin B. Blyuss

Guest Editor

PLOS Computational Biology

Virginia Pitzer

Deputy Editor

PLOS Computational Biology

---

## [Editor Report · Acceptance letter]

16 Jun 2020

PCOMPBIOL-D-19-01812R1 

Computational models to improve surveillance for cassava brown streak disease and minimize yield loss.

Dear Dr Ferris,

I am pleased to inform you that your manuscript has been formally accepted for publication in PLOS Computational Biology. Your manuscript is now with our production department and you will be notified of the publication date in due course.

With kind regards,

Sarah Hammond
